# Unusual features and localization of the membrane kinome of *Trypanosoma brucei*

**Bryan C. Jensen**[1]\*, **Pashmi Vaney**[2¤], **John Flaspohler**[3], **Isabelle Coppens**[4],
**Marilyn Parsons**[1,5]

1 Center for Global Infectious Disease Research, Seattle Children's Research Institute, Seattle, Washington, United States of America, 2 Seattle Biomedical Research Institute, Seattle, Washington, United States of America, 3 Biology Department, Concordia College, Moorhead, Minnesota, United States of America, 4 Department of Molecular Microbiology and Immunology, Johns Hopkins School of Public Health, Baltimore, Maryland, United States of America, 5 Departments of Pediatrics and Global Health, University of Washington, Seattle, Washington, United States of America

¤ Current address: Adaptive Biotechnologies, Seattle, WA, United States of America
\* bryan.jensen@seattlechildrens.org

**Data Availability Statement:** All relevant data are within the manuscript and its Supporting Information files.

## Abstract

In many eukaryotes, multiple protein kinases are situated in the plasma membrane where they respond to extracellular ligands. Ligand binding elicits a signal that is transmitted across the membrane, leading to activation of the cytosolic kinase domain. Humans have over 100 receptor protein kinases. In contrast, our search of the *Trypanosoma brucei* kinome showed that there were only ten protein kinases with predicted transmembrane domains, and unlike other eukaryotic transmembrane kinases, seven are predicted to bear multiple transmembrane domains. Most of the ten kinases, including their transmembrane domains, are conserved in both *Trypanosoma cruzi* and *Leishmania* species. Several possess accessory domains, such as Kelch, nucleotide cyclase, and forkhead-associated domains. Surprisingly, two contain multiple regions with predicted structural similarity to domains in bacterial signaling proteins. A few of the protein kinases have previously been localized to subcellular structures such as endosomes or lipid bodies. We examined the localization of epitope-tagged versions of seven of the predicted transmembrane kinases in *T. brucei* bloodstream forms and show that five localized to the endoplasmic reticulum. The last two kinases are enzymatically active, integral membrane proteins associated with the flagellum, flagellar pocket, or adjacent structures as shown by both fluorescence and immunoelectron microscopy. Thus, these kinases are positioned in structures suggesting participation in signal transduction from the external environment.

## Introduction

With its complex life cycle spanning different hosts and tissues, the response of *Trypanosoma brucei* to its environment is a subject of considerable interest. In other eukaryotes, signaling by extracellular molecules is initiated by ligand binding, typically to proteins situated at the plasma membrane. The most prominent categories of such proteins are G-protein coupled

**Funding:** This work was supported in part by a grant 5R21AI101424 from the National Institutes of Health and by Seattle Children's Research Institute to MP. The funders had no role in study design, data collection and analysis, decision to publish, or preparation of the manuscript.

**Competing interests:** The authors have declared that no competing interests exist.

receptors (absent in trypanosomatids) and protein kinases (PKs) [1]. *T. brucei* also possesses a large family of surface proteins that have external ligand-binding domains and internal adenylate cyclase domains [2], although the ligands and signaling pathways involved remain to be elucidated. Transporters can also function in signal transduction by internalizing regulatory molecules such as Pi or cGMP [3–5]. In *T. brucei*, a plasma membrane carboxylate transporter is important in regulating the transition of mammalian bloodstream forms (BF) to the insect procyclic form (PF) stage [6].

Eukaryotes tend to have hundreds of PKs that bear the signature eukaryotic PK (ePK) motifs in the catalytic domain, *T. brucei* has ~180 protein-coding regions that bear a canonical ePK domain [7]. Almost all PK orthologues are shared among *T. brucei*, *Trypanosoma cruzi* and *Leishmania major*. Most eukaryotes have tens to hundreds of PKs that bear transmembrane domains (TMDs), and in metazoans almost all are tyrosine kinases that reside in the plasma membrane. Ligand binding to the extracellular domain generates a signal that is transduced to the cytosolic catalytic domain resulting in kinase activation. In a previous analysis of the genome, we identified ten PKs predicted to bear TMDs as predicted by TMHMM (transmembrane hidden marker model) [7], The redrawing of CDS boundaries based on 5' transcript mapping eliminated the putative TMD of one (Tb927.10.16160) and annotation of another PK was updated to include a TMD (Tb927.10.1910). The catalytic domains of these PKs cluster with serine/threonine kinases [7].

About half of the *T. brucei* TMD PKs have been characterized functionally. Tb927.2.2720, dubbed *MEKK1* (so-named for the similarity of its catalytic domain to MAP kinase kinase kinases), is required for the quorum sensing pathway that triggers proliferating slender BF to enter the stumpy stationary phase as a prelude to transmission to the insect vector [8]. Tb927.11.14070, known as *RDK1*, encodes a repressor of BF to PF differentiation [9]. Tb927.11.8940 encodes LDK (lipid droplet kinase), which is present on lipid droplets and essential for their biogenesis [10]. Tb927.4.2500, which encodes an eIF2 kinase-related protein, EIF2K2K, is localized to the flagellar pocket [11]. While the function of this kinase in *T. brucei* is not known, its orthologue in *Leishmania* appears to regulate promastigote to amastigote development [12] and the *T. cruzi* orthologue similarly regulates the development of infective epimastigote forms [13]. Tb927.10.1910 is suggested to play a role in melarsoprol sensitivity on the basis of high-throughput RNAi assays [14]. Cell death was not observed upon depletion of the TM kinases in BF in a kinome-wide stem-loop RNAi study (although the degree of knockdown was not quantitated) [9].

Here we report studies examining the TMD PKs. Many of them have unusual domain structures, such as the presence of multiple TMDs, nucleotide cyclase domains, or regions of structural similarity to bacterial signaling modules. We examined the subcellular localization of seven of these PKs in BF single marker clone of strain 427 (this clone is unable to differentiate to proliferative PF). Five of the tagged PKs localized predominantly to the endoplasmic reticulum (ER), indicating they entered the secretory system. Two were found to have particularly distinctive distributions in both BF and PF: FHK localized to a region at or near the flagellar pocket and MEKK1 localized similarly, as well as to the flagellum. Immunoelectron microscopy of cultured PF confirmed the unique positioning of these two kinases.

## Materials and methods

### Parasites

The work described uses PF 29–13 and single marker BF lines, which are derivatives of the *T. brucei* 427 strain [15]. Both lines express T7 RNA polymerase and the tetracycline (Tet) repressor, allowing for Tet-regulated expression of transfected genes. PF were grown in SDM-79

(JRH Biosciences) supplemented with 10% fetal calf serum containing 15 μg/ml G418 and 50 μg/ml hygromycin to maintain the T7 RNA polymerase and Tet repressor genes. BF were grown in HMI-9 with 2.5 μg/ml G418. Plasmids were transfected into PF as described [15] and transfectants were selected with 1 μg/ml puromycin. BF were transfected [16], modified as described [17]. BF transfectants were selected with either 5 μg/ml hygromycin or blastocidin S depending on the plasmid vector.

**Plasmids and cloning.** For expression of V5-tagged proteins in BF, we generated the plasmid pT7-3V5-Hyg by replacing the GFP and lacZ stuffer region of pT7-GFP [18] with the multicloning site and three V5 epitopes derived from pLEW79-3V5-Pac [10]. Expression was driven by a tetracycline (Tet) regulated T7 promoter and the 3' UTR was derived from aldolase. For expression of epitope tagged proteins in PF (V5 or HA), coding sequences were cloned into either pLEW79-3V5-PAC or plasmid pLEW100v5-3HA-BSD. The latter was created by replacing the luciferase gene in pLEW100v5-BSD with the multicloning sites derived from pLEW79-3V5-PAC and three HA epitope tags. Expression was driven by a tetracycline (Tet) regulated PARP promoter and the 3' UTR was derived from aldolase. The coding sequences for the PKs were amplified and PCR products were digested with the restriction enzymes noted in the primer and ligated into appropriately digested vectors. The expression construct for Tb927.9.3120 also included the endogenous splice acceptor. All primers are described in S1 Table.

**Western blots.** Cell lysates were generated, and proteins resolved by SDS-PAGE and transferred to nitrocellulose as described [19]. V5-tagged proteins were detected with mouse monoclonal anti-V5 antibody (ThermoFisher) at 0.3 μg/ml. HA-tagged proteins were detected with mouse (Covance) or rat (Roche) monoclonal anti-V5 antibodies at 1 μg/ml or 12.5 ng/ml respectively. Vacuolar phosphatase [20] was detected with anti-VTC at 1:5000. Bound antibodies were revealed with either IRDye 800CW dye-conjugated goat anti-mouse Ig and IRDye680RD dye-conjugated goat anti-rabbit Ig (LI-COR) at 25 ng/ml and data was visualized on a LI-COR Odyssey.

**Immunofluorescence analysis.** IFAs were performed as previously described [19]. In brief, BF parasites were pelleted and washed with PSG (PBS with 10 mM glucose), then pelleted and fixed in 4% paraformaldehyde in PBS for 5 min prior to placing on poly-L-lysine coverslips. PF IFAs were performed as described for BF cells with the exception that coverslips were not treated with poly-L-lysine. These fixation conditions have been previously used to localize membrane proteins in both BF and PF trypanosomes [21–24]. The V5 epitope tag was detected by using mouse monoclonal anti-V5 antibody (Invitrogen) at 1 ug/ml, followed by goat anti-mouse IgG conjugated with fluorescein isothiocyanate (FITC) (Southern Biotechnology). Additional antibodies used for colocalization included antibodies directed against BiP [25] and PIFTC3 [26], used at 1:1000; the cation channel TcCat [27], used at 1:250; YL1/2 (Gene-Tex), used at 1 μg/ml. To visualize the flagellar pocket, pre-cooled cells were incubated at 4˚C with biotinylated tomato lectin (Vector Laboratories) at 2 μg/ml for 30 minutes. Cells were then pelleted washed with chilled PSG and fixed and processed for IFA as described above. Tomato lectin was visualized with 2 μg/ml streptavidin conjugated to either Texas Red or Alexa Fluor 488 (ThermoFisher). Dye-conjugated secondary antibodies (Southern Biotechnology) were used at 2 μg/ml. Images were scaled to span the graph of signal intensity of the image field. The Pearson's coefficient for colocalization was measured using Imaris software (Oxford Instruments). Individual parasites were identified using the signal for the V5-tagged PKs. Voxels (three dimensional pixels) with signal above background for the tagged proteins were used for analysis against all voxels in the channel used to detect BiP.

**Immunoelectron microscopy.** Preparations of parasites were fixed in 4% paraformaldehyde (Electron Microscopy Sciences, PA) in 0.25 M HEPES (pH 7.4) for 1 h at room

temperature, then in 8% paraformaldehyde in the same buffer overnight at 4˚C. They were infiltrated, frozen and sectioned as previously described [28]. The sections were immunolabeled with high affinity rat anti-HA monoclonal antibody 3F10 (1:50 in PBS/1% fish skin gelatin), then with anti-rat IgG antibodies, followed directly by 10 nm protein A-gold particles (Department of Cell Biology, Medical School, Utrecht University, the Netherlands) before examination with a Philips CM120 Electron Microscope (Eindhoven, the Netherlands) under 80 kV.

**Cell fractionation and kinase assays.** PF expressing HA-tagged proteins washed in PBS containing 10 mM glucose. They were permeabilized with digitonin and the organellar pellet was extracted with sodium carbonate as described [29] except that Complete Protease tablets (Roche) were included for both digitonin permeabilization and carbonate extraction, to reduce protein degradation. Cell lysates from PF cells expressing either V5-tagged MEKK1 or HA-tagged FHK along with the untransfected parental line 29–13 were prepared as described [30]. Epitope-tagged proteins were immunoprecipitated from cell lysates prepared from $10^8$ cells. Kinase assays were done as described [31] using $^{32}$PγATP with 5 μg myelin basic protein as an exogenous substrate. Reactions were resolved by SDS-PAGE and transferred to nitrocellulose and labeled proteins detected by phosphorimaging. Blots were subsequently probed with antibodies directed against either V5 or HA.

## Results

### Domain structure

Fig 1 depicts schematics of the ten *T. brucei* PKs that have TMDs predicted by TMHMM 2.0 [32], showing the location of the TMDs. Cross-validation of this algorithm with proteins of known topology showed that both over-prediction of TM helices and under-prediction are approximately 2.5%. For eukaryotic proteins, the authors showed that the most common cause overprediction is confounding of a signal peptide with a TMD [32]. Also shown are the PK catalytic domains and additional domains listed on TriTrypDB (based primarily on sequence similarities), plus regions HHpred [33] identified as having predicted structural similarities to other protein modules. These PKs are large, with the smallest (the lipid droplet kinase LDK) being 553 aa, about twice the size of the PK domain, and the rest ranging from 929 to 1678 aa. The PK domain of all of these kinases, with the exception of LDK, resides at the C-terminus.

### Transmembrane domains and signal peptides

Surprisingly, seven of the ten proteins are predicted by TMHMM to be multi-pass proteins, having two or more TMDs, a topology that is extremely uncommon for eukaryotic PKs. We therefore re-examined the TMDs (predicted by TMHMM and shown TriTrypDB, red bars in Fig 1) using CCTOP (Constrained Consensus TOPology Prediction, http://cctop.enzim.ttk. mta.hu/), which combines ten algorithms with structural data to yield a model [34]. The CCTOP models concurred with TMHMM for four PKs and proposed additional TMDs for the rest of the PKs (Fig 1, red arrowheads), and rejected one of the TMDs on RDK1 (red x). Some of the additional TMDs predicted with CCTOP are unlikely to be authentic since they occur in the middle of domains: one that punctuates the ePK catalytic domain in Tb927.5.3150, and two each within structural domains of RDK1 and Tb927.9.3120. When we examined the orthologues of these *T. brucei* PKs in *L. major* and *T. cruzi*, the number and position of the TMHMM-predicted TMDs generally concurred (S2 Table). The exceptions included the *L. major* orthologue of eIF2K2 (LmjF.34.2150), the *T. cruzi* CL-Brener orthologues of Tb927.5.3150, and the *T. cruzi* orthologue of LDK (TcCLB.511801.14), all of which lack a TMHMM-predicted TMD. However, CCTOP did predict TMDs in these proteins.

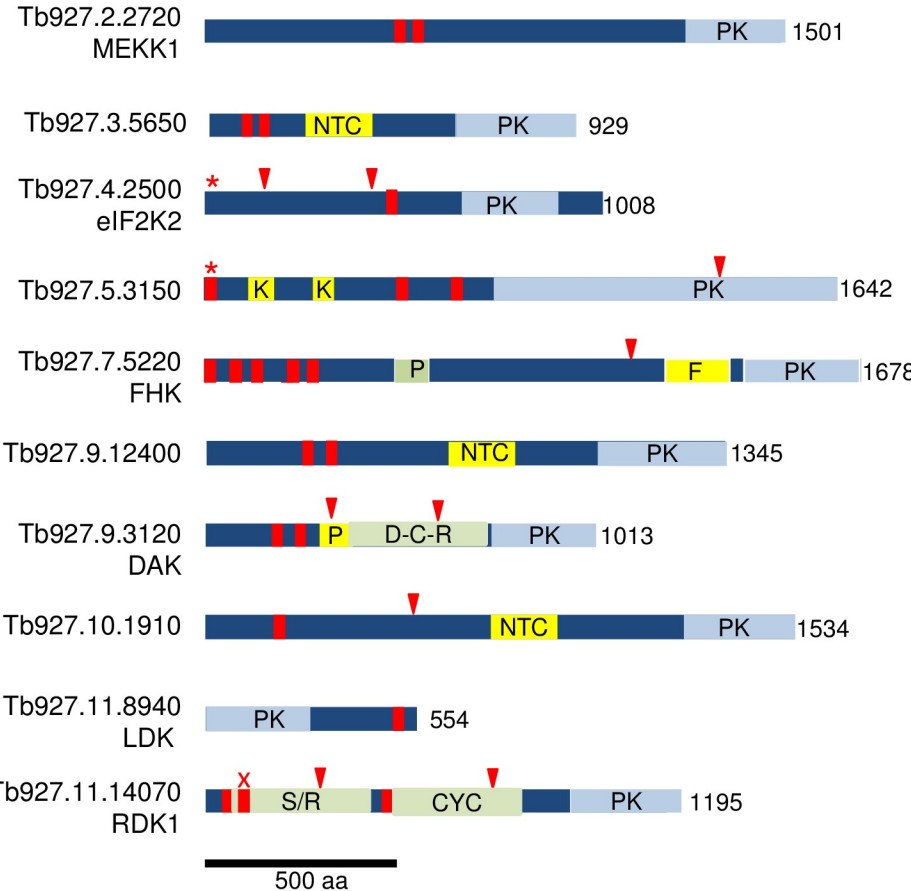

**Fig 1. Schematics of *T. brucei* protein kinases predicted to bear transmembrane domains.** TMDs predicted by both TMHMM and CCTOP are marked with red bars (one predicted by TMHMM only is marked with a red X); additional TMDs predicted by CCTOP are marked by red arrowheads. Predicted signal sequences are indicated by a red asterisk. Light blue regions are ePK Pfam domains. Yellow bars depict other domains: PAS motif (P), forkhead associated domain (F), nucleotide cyclase-like domain (NTC), Kelch propeller blade (K). Light green bars demarcate regions with predicted structural similarity to known structures revealed by HHpred. Bacterial DHp, catalytic, and receiver domains characteristic of histidine kinases (D-C-R) in Tb927.9.3120, as well as bacterial sensor/receptor domains (S/R) and adenylyl/guanylyl cyclases (AGC) in RDK1, exhibited probabilities >95% (see Fig 2 and S2 Fig for details). The PAS-like domain in FHK was detected with >90% probability. Length of the predicted protein in amino acids is indicated.

Furthermore, since LDK resides in a monolayer membrane surrounding lipid droplets rather than a bilayer, algorithms may be less proficient at identifying relevant membrane insertion/association sequences. While it is not possible to be certain of the number of TMDs from bioinformatic predictions, an even number of TMDs would place the N- terminal region and C-terminal catalytic domain in the same compartment. For example, MEKK1 is predicted to have a hairpin of two TMDs, separated by only four aa. The extended N- and C-regions would either both be cytosolic, or both be extracellular. We are unaware of any transmembrane

kinases that display catalytic domains on the external face of the plasma membrane, so propose that the former possibility is most likely.

Most membrane proteins destined for the secretory system or plasma membrane enter the ER membrane co-translationally, courtesy of a signal peptide or a TMD [35]. Interestingly, it has been proposed that two TMDs separated by loop, such as seen in MEKK1 and several other TMD kinases (Fig 1), may be a topogenic signal for insertion into the ER membrane [36]. Two of the PKs bearing TMDs are also predicted to bear signal peptides: eIF2K2 and Tb927.5.3150. Three additional *T. brucei* PKs not examined here are annotated as bearing signal peptides, but they lack predicted TMDs.

## Domains identified by sequence similarity

Searches based on sequence similarity showed that Tb927.5.3150 bears two Kelch/Kelch-like domains, which often participate in protein-protein interactions. Additional domains related to signaling functions were detected on several PKs. Three bear nucleotide cyclase-like domains (Tb927.3.5650, Tb927.9.12400, and Tb927.10.1910). Interestingly these are the only *T. brucei* PKs identified as having such domains, although the parasite genome encodes multiple adenylyl cyclases with cytoplasmic catalytic domains and presumed ligand-binding extracellular domains [37]. One PK (Tb927.9.3120) contains a region related to PAS (Per-Arnt-Sim) domains, which usually function to bind small molecules or proteins intracellularly. Forkhead kinase, FHK, Tb927.7.5220 is so-named for the presence of a forkhead-associated domain; these domains typically interact with phosphopeptides (usually phosphothreonine in specific contexts) [38].

## Conserved folds identified by structural predictions

The PKs were also subjected to HHpred analysis to identify regions with predicted structural similarity to protein structures available in the protein database PDB [33] (Fig 1). The domains mentioned above all showed numerous matches with the related domains on other proteins, with probability scores of over 99%. A PAS-like region on FHK was additionally identified by HHpred, with multiple hits having a probability above 90%.

Surprisingly, two PKs have large regions of predicted structural similarity to molecular signaling modules present in bacteria. RDK1 includes two regions with predicted structural similarity to other signaling molecules (S1 Fig). The first is similar to bacterial sensor/receptor domains, such the ligand binding domain of *Pseudomonas aeruginosa* histamine receptor TlpQ [39] and the extracellular domains of a set of bacterial histidine kinases for which the ligands are unknown [40]. This region is followed by a predicted TMD and then a segment predicted to fold similarly to full-length catalytic domains of several bacterial and eukaryotic adenylyl/guanylyl cyclases. The cyclase region is followed by the eukaryotic PK catalytic domain. This organization suggests that the sensor-like domain lies on the opposite side of the membrane to the cyclase and PK domains. The modular organization of RDK1 is conserved in other trypanosomatids, as well as the more distantly related free-living bodonid *Bodo saltans* [41].

Tb927.9.3120 has co-linear regions predicted to fold similarly to molecules involved in bacterial signaling: histidine kinases and response regulators. The histidine kinases typically contain sensing modules, followed by DHp dimerization regions, then catalytic/ATP binding domains (which phosphorylate a histidine on DHp). Response regulators include receiver domains (which receive the phosphate from DHp) and effector domains. In some cases, the kinase and receiver reside on the same molecule (hybrid histidine kinases), although two component systems are more common. Importantly, in Tb927.9.3120 the regions corresponding

to each domain (as defined in [42, 43]) are essentiall complete (see Fig 2). HHPRED analysis shows that this organization of a hybrid histidine kinase cassette followed by a eukaryotic protein kinase domain is conserved in the Tb927.9.3120 orthologues in other trypanosomatids, such as *T. cruzi*, *L. major*, and *Bodo saltans* (Uniprot numbers A0A2V2UQT0, Q4Q920 andA0A0S4IKM7 respectively). Together, this provides intriguing evidence of a bacterial genetic incursion of a signaling cassette into a common ancestor of bodonids and

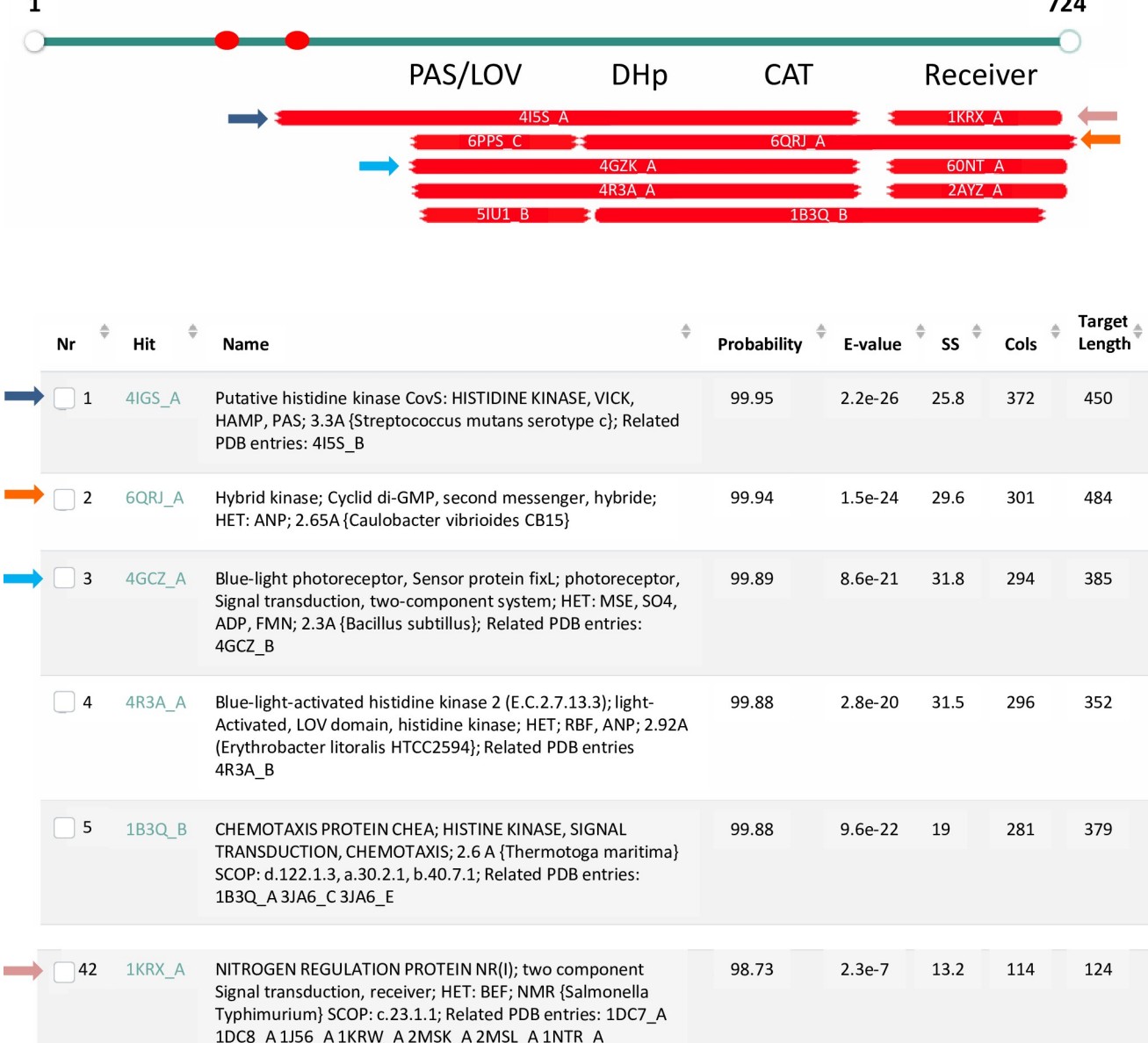

| Nr | Hit | Name | Probability | E-value | SS | Cols | Target Length |
|----|-----|------|-------------|---------|-----|------|---------------|
| 1 | 4IGS_A | Putative histidine kinase CovS: HISTIDINE KINASE, VICK, HAMP, PAS; 3.3A {Streptococcus mutans serotype c}; Related PDB entries: 4I5S_B | 99.95 | 2.2e-26 | 25.8 | 372 | 450 |
| 2 | 6QRJ_A | Hybrid kinase; Cyclid di-GMP, second messenger, hybride; HET: ANP; 2.65A {Caulobacter vibrioides CB15} | 99.94 | 1.5e-24 | 29.6 | 301 | 484 |
| 3 | 4GCZ_A | Blue-light photoreceptor, Sensor protein fixL; photoreceptor, Signal transduction, two-component system; HET: MSE, SO4, ADP, FMN; 2.3A {Bacillus subtillus}; Related PDB entries: 4GCZ_B | 99.89 | 8.6e-21 | 31.8 | 294 | 385 |
| 4 | 4R3A_A | Blue-light-activated histidine kinase 2 (E.C.2.7.13.3); light-Activated, LOV domain, histidine kinase; HET: RBF, ANP; 2.92A (Erythrobacter litoralis HTCC2594); Related PDB entries 4R3A_B | 99.88 | 2.8e-20 | 31.5 | 296 | 352 |
| 5 | 1B3Q_B | CHEMOTAXIS PROTEIN CHEA; HISTINE KINASE, SIGNAL TRANSDUCTION, CHEMOTAXIS; 2.6 A {Thermotoga maritima} SCOP: d.122.1.3, a.30.2.1, b.40.7.1; Related PDB entries: 1B3Q_A 3JA6_C 3JA6_E | 99.88 | 9.6e-22 | 19 | 281 | 379 |
| 42 | 1KRX_A | NITROGEN REGULATION PROTEIN NR(I); two component Signal transduction, receiver; HET: BEF; NMR {Salmonella Typhimurium} SCOP: c.23.1.1; Related PDB entries: 1DC7_A 1DC8_A 1J56_A 1KRW_A 2MSK_A 2MSL_A 1NTR_A | 98.73 | 2.3e-7 | 13.2 | 114 | 124 |

**Fig 2. HHpred analysis of Tb927.9.3120 shows evidence of ancient domains.** The PK domain (aa 737–980) is not shown. Red ovals–transmembrane domains predicted by TMHMM and CCTOP. Each bar below the gene model represents the top hits by HHpred. The color of the arrows mark descriptions matching the hits. The description includes the PDB number and chain designation (Hit), a description of the PDB entry including related PDB entries (Name), the probability of the hit based on the Hidden Markov Model (Probability), the probability of the match in an unrelated database (E-value), score for the secondary structure prediction (SS), number of amino acids aligned (Col), and the total length of the target in PDB (Target Length). The documentation for HHpred considers "Probability" the most important criterion with our hits meeting at least one of the two most stringent criteria: having a score >95% or having a score >50% and making reasonable biological sense [33].

trypanosomatids. Accordingly, we propose to name this gene dual ancestor kinase (*DAK*). At this time no phenotypes have been associated with knockdown of Tb927.9.3120, making it difficult to probe the roles of its various domains.

## Expression

We examined the expression of these PKs using our previous genome-wide ribosome profiling data from pleiomorphic slender form BF (isolated from mice) and cultured PF [44]. Ribosome profiling provides a direct measure of protein production. In Fig 3, the dark blue and red bars show the abundance of ribosome-associated mRNA footprints of BF and PF respectively. For comparison, the mRNA abundances are also shown (light blue and pink bars). Several of the PKs show higher protein production in slender BF than in PF, with MEKK1 showing 3-fold more expression in slender BF and eIF2Ka having 2-fold more. However, the most dramatic difference was for RDK1 with 7-fold more mRNA and 40-fold more protein production in slender BF than in PF. Conversely, DAK had about twice as much protein production in PF than slender BF, although the mRNA levels are very similar between stages. Most of the PKs were close to the median level of protein production for all non-pseudogenes (224 reads per kb in BF and 202 in PF). However, three PKs showed lower protein production, being in the lowest quartile of all PKs. LDK protein production was low in both stages; FHK protein production was low in BF and very low in PF; and Tb927.9.12400 protein production was almost absent in BF and very low in PF.

## Subcellular localization

Previous work localized *T. brucei* LDK to the lipid droplet membrane [10] and eIF2K2 to the flagellar pocket [11] (and its *T. cruzi* orthologue to endosomes [13]). Additionally, the protein

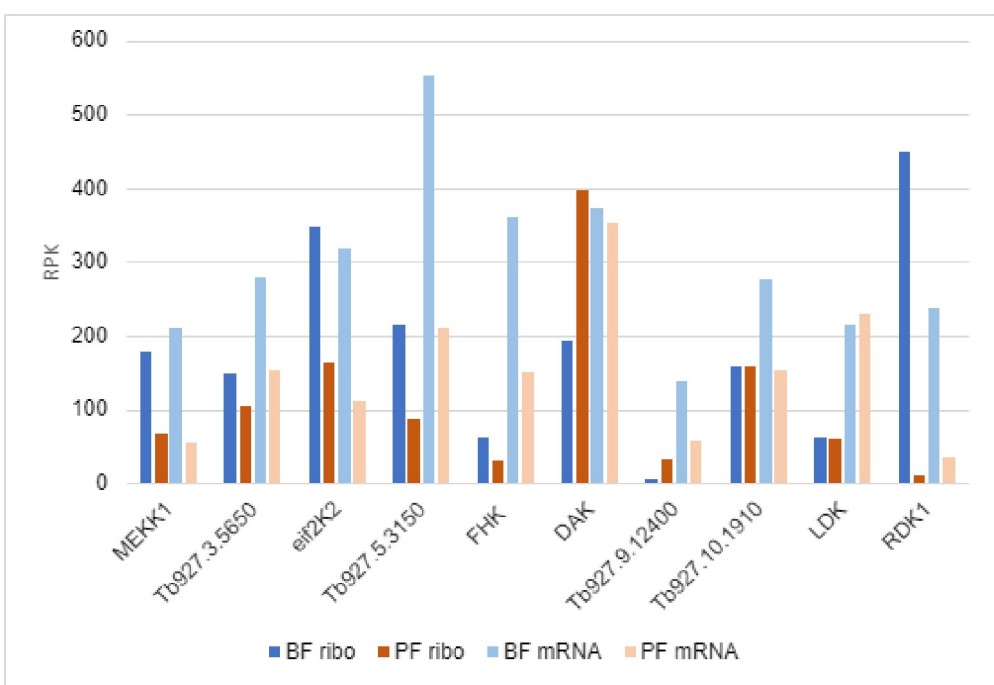

**Fig 3. Protein production and mRNA levels of TMD PKs.** Protein production was measured by ribosome profiling and mRNA levels were measured in the same biological samples (three replicates each). Data is shown as reads per kb and is derived from [44].

encoded by Tb927.10.1910 was localized to the cytoplasm and endocytic organelles in PF in the TrypTag project [45]. We determined the localization of C-terminally tagged versions of the remaining transmembrane PKs in single marker strain BF. No detrimental effects were observed in the transfectants, which utilized a strong promoter to ensure detection of the tagged proteins. A western blot (S2 Fig) showed that all proteins were expressed. The tagged proteins migrated as if they were 10–20% larger than their predicted sizes (see S2 Table), except for Tb927.9.3120 which migrated true to size. Membrane proteins are known to exhibit anomalous migration on SDS-PAGE gels, often migrating either slower or faster than predicted, with apparent molecular masses [46]. In the absence of antibodies specific to individual PKs, it is not possible to predict whether the expression levels of the tagged protein are comparable to the endogenous levels. However, the very low levels of FHK and Tb927.9.12400 revealed by ribosome profiling compared to their ready detection by western blot (Fig 3) suggest that the tagged versions of those PKs are most likely to be over-expressed.

**Many C-terminally tagged TMD PKs localize to endomembranes.** None of the tested PKs is classified as orthologous to the mammalian ER kinases PERK or IRE1 [7]. However, as summarized in Table 1 and shown in Fig 4, five of the tagged PKs colocalized with the ER marker BiP. Staining of the tagged proteins is evident around the body of the cell but, like BiP, it does not extend along the flagellum after it passes the anterior end of the cell body. As with BiP, some perinuclear staining is also seen; in most cases this is fainter than the peripheral staining. RDK1 was previously shown to be associated with membranes and localized to the periphery of the parasite [9]; our colocalization studies indicate that this peripheral localization

**Table 1. Summary of localization studies and function.**

| GeneID/name | TrypTag<br><br>PF[a], endogenous | other studies | This study<br><br>BF, ectopic,<br><br>C-terminal tag | Function[c] |
|---|---|---|---|---|
| Tb927.2.2720<br><br>MEKK1 | N -faint[b], cytoplasm and flagellum; C—faint, cytoplasmic, reticular | | Flagellum, flagellar pocket; similar in PF | Slender to stumpy BF transition [8] |
| Tb927.3.5650 | nd[d] | | ER | nd |
| Tb927.4.2500<br><br>eIF2K2 | N—very faint, cytoplasmic puncta; C—nd | Flagellar pocket and endosomes [11] | nd | Life cycle transitions in *T. cruzi* [13] and *Leishmania* [12] |
| Tb927.5.3150 | N- nd; C–reticulated with occasional bright puncta | | ER | |
| Tb927.7.5220<br><br>FHK | nd | | At or near flagellar pocket, similar in PF | nd |
| Tb927.9.3120 | nd | | ER | nd |
| Tb927.9.12400 | nd | | ER | nd |
| Tb927.10.1910 | N—faint, endocytic and cytoplasm; C -ND | | nd | Melarsoprol sensitivity [14] |
| Tb927.11.8940<br><br>LDK | nd | Lipid droplets [10] | nd | Biogenesis of lipid bodies [10] |
| Tb927.11.14070<br><br>RDK1 | nd | Periphery [9] | ER | Repressor of BF to PF development [9] |

[a] TrypTag: in situ tags the test protein with neon-green at the N- terminus (N) or C-terminus (C) of the protein for expression in PF [45].

[b]Faint, 10-20th percentile of all lines expressing tagged proteins; very faint, <10th percentile. Note that these are all fainter than the parental line, but in cases where proteins are localized to specific organelles, localization can often be visualized.

[c]Function/pathway in *T. brucei* unless otherwise noted.

[d]nd, not determined.

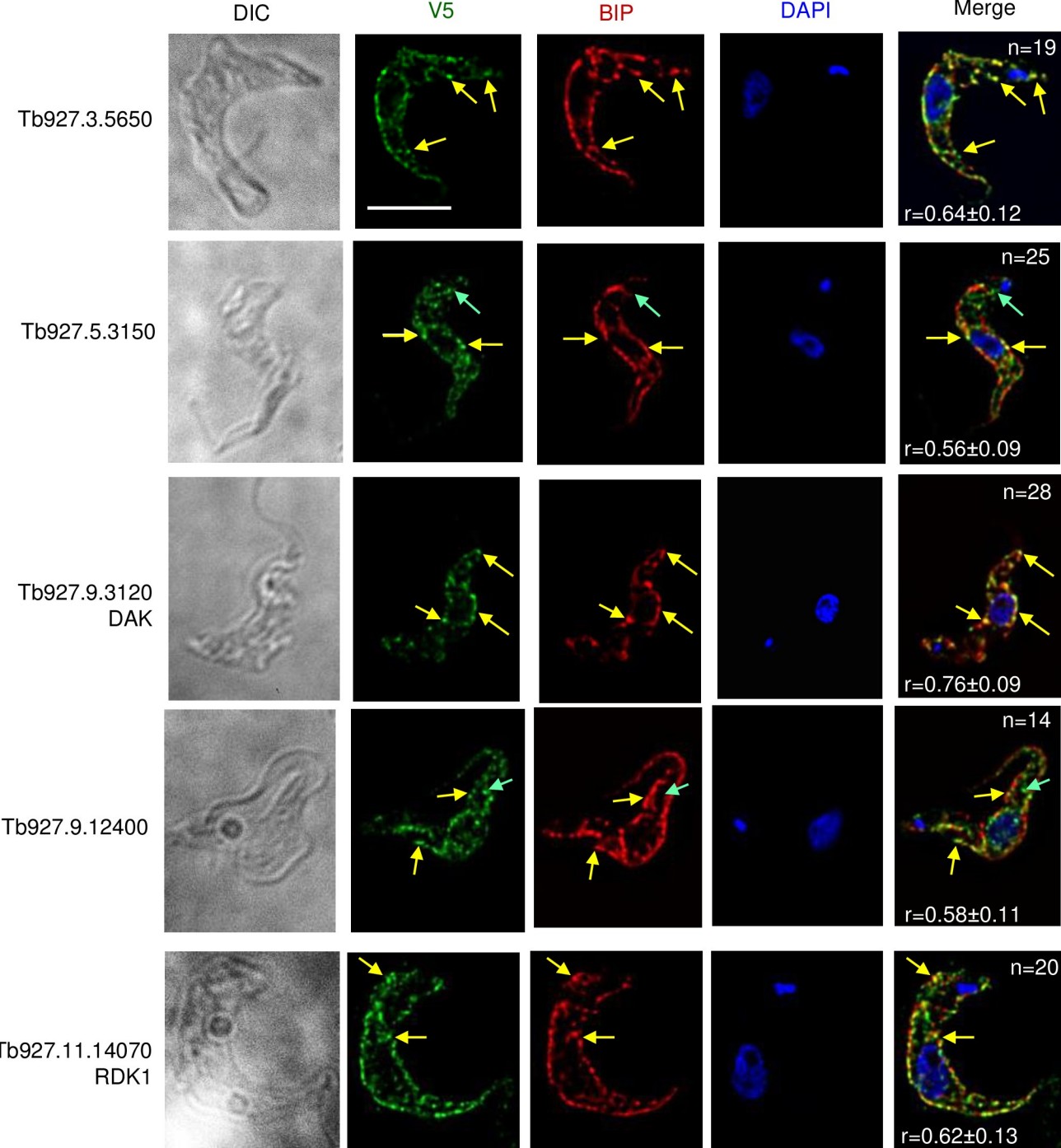

**Fig 4. Localization of V5-tagged predicted membrane kinases to the ER in bloodstream form *T. brucei*.** BF were fixed and stained with primary antibodies as described in Materials and Methods. Bound antibodies were revealed with secondary fluorochrome-coupled antibodies. Green, V5-tagged proteins, Red, anti-BiP, Blue, DAPI; Merge, all three colors. Each series shows a single deconvolved plane, except the DIC image, which is a projection. All images are shown at the same magnification. The Pearson correlation coefficients of the V5-tagged PK and BiP signals are shown in the merged image, along with the number of cells analyzed for each PK. Yellow arrows mark examples of regions of colocalization. For Tb927.5.3150 and Tb927.9.12400, green arrows marks regions in the posterior where the V5 does not co-localize with BiP. Bar, 5 μm.

is the ER. In the case of Tb927.5.3150, we also observed some V5 staining that did not colocalize with BiP at the posterior end of the cells, This region is known to be enriched for Golgi and endocytic organelles see [47], The visual inspection was confirmed by measuring the Pearson's correlation coefficients, which ranged from 0.56 and 0.76 for BiP and the tagged PKs. As expected, the correlation coefficients between DAPI and the tagged PKs Pearson's (0.05–0.11) showed no evidence of co-localization.

**MEKK1 is an integral membrane protein of the flagellum and flagellar pocket with kinase activity.** Immunofluorescence analysis showed that MEKK1-V5 is present along the flagellum, where it aligns with PIFTC3, a component of the flagellar protein transport system [26] and is clearly distinct from the flagellar K+ channel that also extends along the flagellum and localizes to the flagellar attachment zone in *T. cruzi* [27] (Fig 5A and 5B). We also observed staining adjacent to the kinetoplast, which was distinct from, but adjacent to, the depot of PIFTC3 and the basal body as revealed by antibody YL/2 (Fig 5C). In this image it is also very clear that staining extends past the cell body (and flagellar attachment zone) to the tip of the flagellum. When two kinetoplasts were seen, two puncta of both PIFTC3 and MEKK1 were visible. The presence of protein at the base of the flagellum is consistent with localization to the flagellar pocket. To verify this localization live BF expressing MEKK1-HA were incubated with biotinylated tomato lectin at 4° C and then fixed for anti-HA staining (Fig 5D). The low temperature allows lectin binding to the flagellar pocket without internalization into the parasite. Some of the tomato lectin staining overlapped the large puncta of MEKK1. Similar localization of C-terminally tagged MEKK1 was seen in PF cells (Fig 5D), but with some additional staining for both HA-tagged MEKK1 and PIFTC3 consistent with the ER. Immunoelectron microscopy was performed using PF expressing MEKK1 C-terminally tagged with HA epitopes in combination with a high affinity HA antibody that is superior for immunoelectron microscopy. As shown in Fig 6, gold particle staining was present at the flagellar pocket (A, B) but not at the kinetoplast (C). In a dividing cell, both flagellar pockets were stained. Note that the gold particles are predominantly on the cytoplasmic face of the flagellar pocket, as expected. MEKK1-HA was also seen along the flagellum (Fig 6E). The MEKK1-HA at the flagellar pocket may represent molecules in transit to the flagellum or a final destination for a subset of MEKK1.

We next assessed kinase activity and membrane association of MEKK1. PF were used for these experiments (and for similar experiments assessing FHK, see below) due to the ease of obtaining the larger number of cells required for the experiments, and is justified since the tagged protein showed similar localization in BF and PF cells. To assess membrane association, we conducted cell extraction and fractionation of PF expressing MEKK1-HA. The full-length protein migrated anomalously, somewhat faster than the 220 kDa marker (the tagged protein is predicted to be 167 kDa). The ~95 kDa V5-tagged fragment can be disregarded since it is not seen in cells lysed directly in SDS sample buffer, and therefore represents a degradation fragment. As shown in the immunoblot in Fig 7, upon treatment of PF with digitonin to release the cytosol, the intact kinase remained in the pellet fraction (P1). Subsequent treatment of the digitonin pellet with sodium carbonate, pH 11, showed MEKK-V5 in the pellet (P2), demonstrating that it is an integral membrane protein. Marker proteins PGKB (cytosolic), PGKA (glycosomal which shows partial membrane association) [48] and vacuolar pyrophosphatase, TcVSP) (integral vacuolar membrane) [49] verified the fractionation procedure.

To test for catalytic activity, anti-V5 immunoprecipitates from PF expressing MEKK1-V5, as well as from the untransfected parental 29–13 line, were incubated with the exogenous substrate myelin basic protein (MBP) in the presence of $\gamma^{32}$P-ATP. The proteins were then separated by SDS-PAGE and analyzed by phosphorimaging (Fig 7B, top) and western blot (Fig 7B, bottom). MBP was phosphorylated in the sample containing MEKK1-V5, but not the

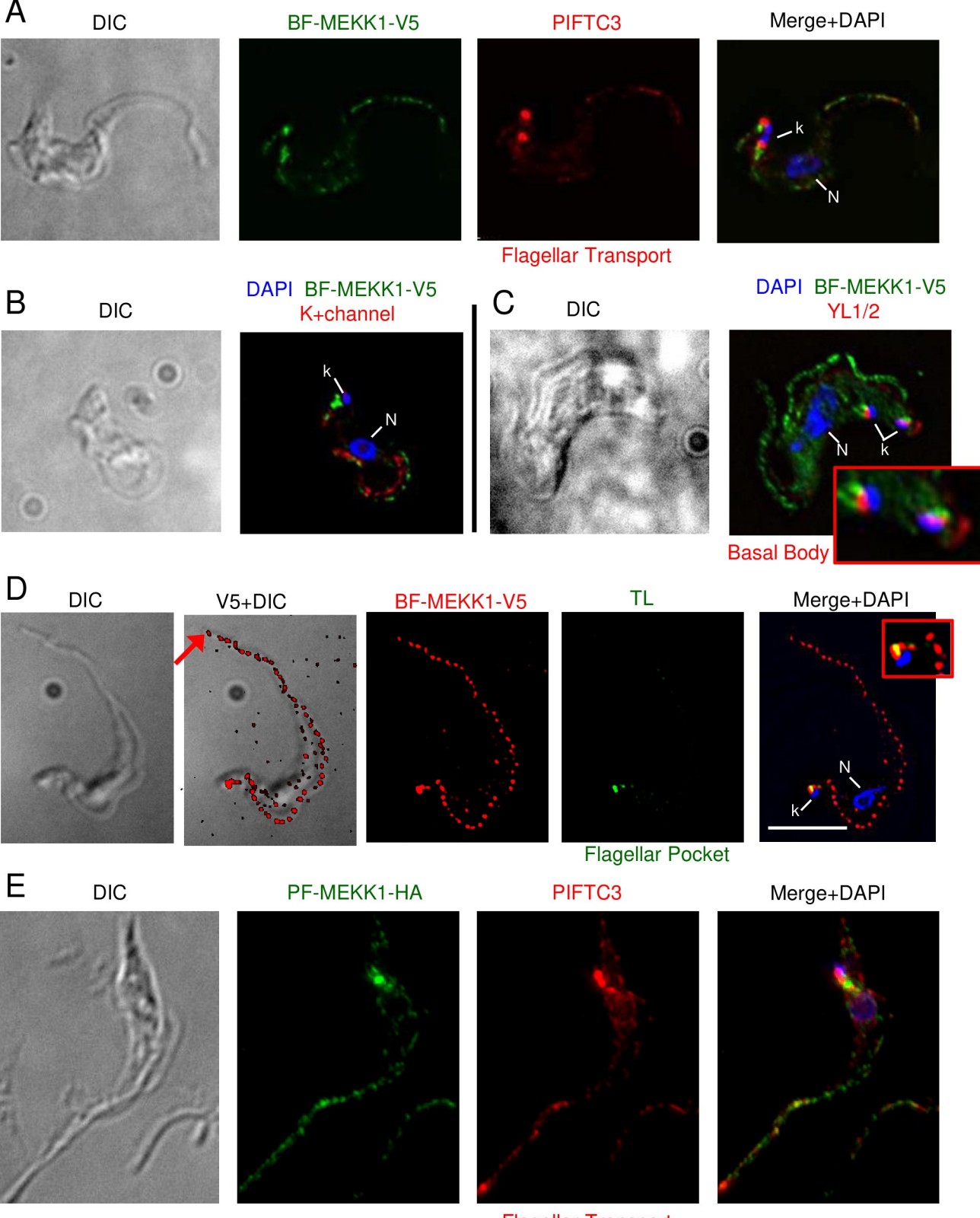

**Fig 5. Immunofluorescence localization of MEKK1-V5 (Tb927.2.2720) in *T. brucei*.** A) Co-staining of BF with antibody *to T. brucei* PIFCT3 (red) [26], a component of the flagellar transport system. B) Co-staining of BF with antibody to a K+ channel protein that colocalizes with the flagellar attachment

zone in *T. cruzi* (left, red) [27] <u>C)</u> Co-staining with monoclonal antibody YL1/2 (right, red) which detects basal bodies. The image shows a cell with two kinetoplasts that appears to be in early mitosis. <u>D)</u> co-staining of BF with tomato lectin (TL, green) at 4˚, which marks the flagellar pocket. <u>E)</u> Co-staining of MEKK1 with PIFTC3 cells showing similar localization. Each series shows a single deconvolved plane, except DIC images, which are projection. All images, except enlargements, are shown at the same magnification (bar, 5 μm). Enlargements show detail near the kDNA. N, nucleus; k, kDNA.

untransfected control, thereby demonstrating that MEKK1 can phosphorylate an exogenous substrate. It is formally possible, albeit unlikely, that the tagged kinase is inactive but interacts with an active kinase responsible for the detected activity.

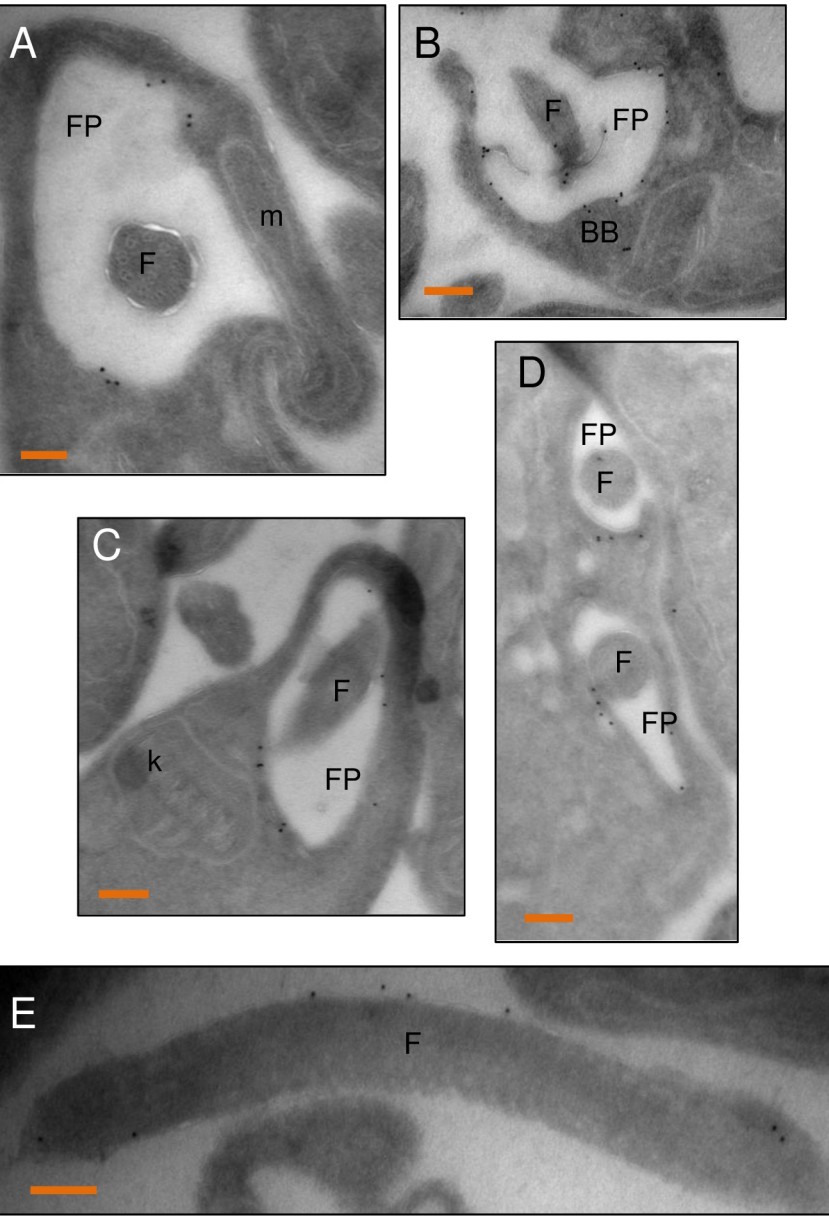

**Fig 6. Localization of MEKK1-HA to the flagellum and flagellar pocket of procyclic forms by immunoelectron microscopy.** PF expressing MEKK1 C-terminally tagged with HA epitopes were stained with anti-HA, and staining revealed by protein-A gold 10 nm particles. A) The gold staining is on the flagellar pocket (FP) membrane, flagellum (F), but absent from the mitochondrion (m). B) staining is observed at the base of the flagellar pocket at the basal bodies (BB). C) no staining of the kinetoplast (k) is seen. D) in a dividing cell, both flagellar pockets are stained. E) the external portion of the flagellum is also stained. Bars, 100 nm.

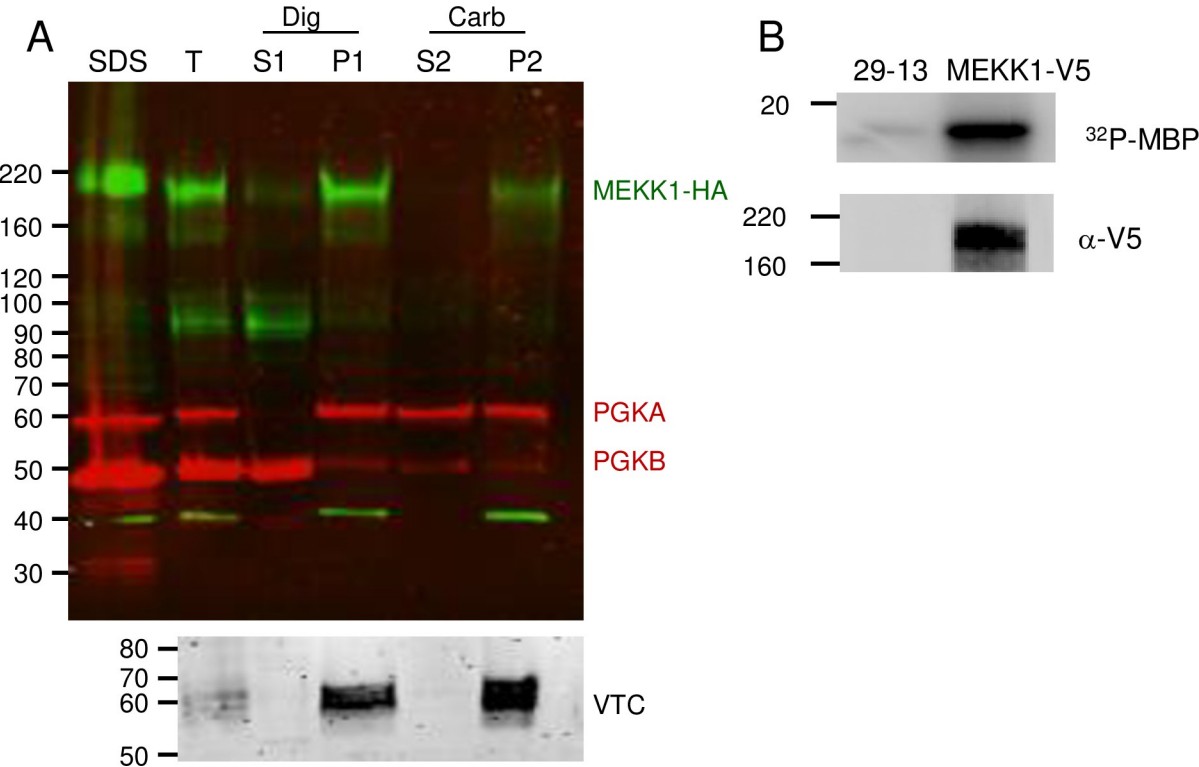

**Fig 7. MEKK1 is an integral membrane protein which has protein kinase activity. A)** Cell fractionation. PF cells expressing MEKK1-HA were fractionated by digitonin treatment followed by carbonate extraction. After separation by SDS-PAGE and blotting, the blots were probed with mouse anti-HA (green) and rabbit anti-phosphoglycerate kinase (red) which detects the 56 kDa glycosomal form (PGKA) and the 47 kDa cytosolic isoform (PGKB). A parallel blot was also incubated with antibody to the acidocalcisome membrane protein vacuolar phosphatase (VTC). SDS, SDS lysate; T, total after incubation with digitonin; S1, digitonin supernatant; P1, digitonin pellet; S2, carbonate supernatant; P2, carbonate pellet. Lanes contain fractions from $3 \times 10^6$ cells. **B)** Kinase activity. Immunoprecipitated MEKK1 from PF was incubated with $\gamma$ $^{32}$P-ATP and the exogenous substrate myelin basic protein (MBP), followed by SDS-PAGE and transfer to nitrocellulose. A control immunoprecipitation used the untransfected parental line (29–13). Top: phosphorimaging demonstrates phosphorylation of MBP. Bottom: the blot was probed with anti-V5 mAb demonstrating the pulldown of the tagged protein.

**FHK is an integral membrane protein of the flagellar pocket with kinase activity.** Like MEKK1, FHK-V5 demonstrated staining near the kinetoplast, close to PIFTC3 upon immuno-fluorescence analysis of BF (Fig 8A). However, staining did not extend along the flagellum and little co-localization with BiP was observed (Fig 8B). The HA staining was adjacent to and possibly overlapping the tomato lectin signal (Fig 8C). Individual deconvolved planes showed FHK between the nucleus and kinetoplast in what appears to be a basket-like structure (Fig 8D). Fig 8E depicts a PF parasite stained for DNA and FHK. In BF and PF, FHK staining retained its adjacency to the kinetoplast, despite the different locations of the kinetoplast in the two stages (posterior in BF, near the nucleus in PF). Immunoelectron microscopy of PF expressing FHK-HA demonstrated that the protein was localized to the flagellar pocket (with gold particles being cytoplasmic rather than luminal) but not to the flagellum or kinetoplast (Fig 9A, enlarged in 9A). Occasionally staining was seen more prominently on one side of the flagellar pocket (Fig 9B, enlarged in 9B). Sometimes staining was observed on vesicles close to the flagellar pocket, but only sporadically outside the region of the pocket (Fig 9C and 9D).

PF expressing FHK-HA were fractionated as described above for MEKK1. Full-length FHK-HA was in the digitonin pellet, and the carbonate-insoluble fraction, indicating it is an integral membrane protein (Fig 10A). As with MEKK1, the smaller degradation fragment can be disregarded since it was absent from cells lysed in SDS sample buffer. To test for catalytic

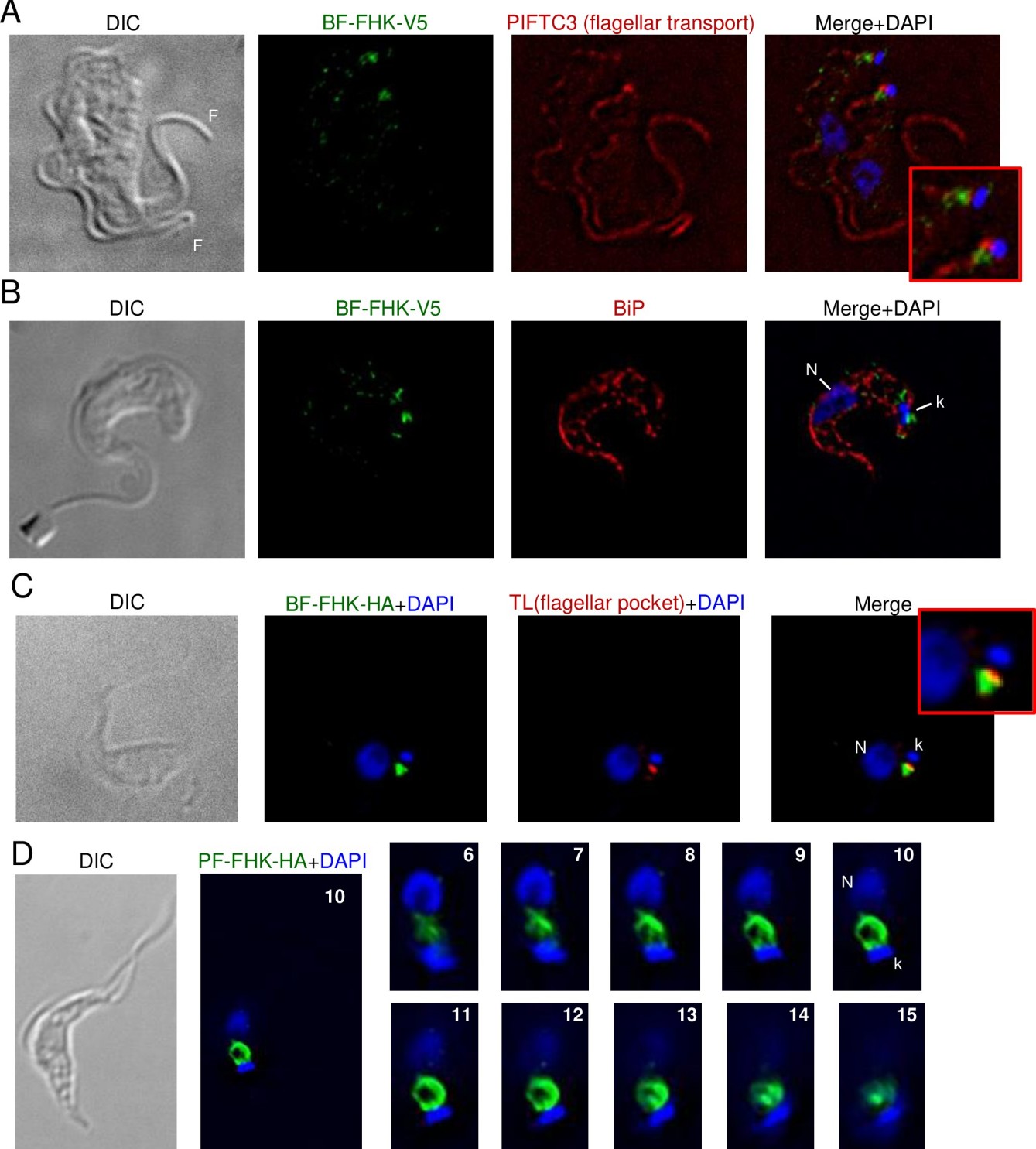

**Fig 8. Predominant localization of epitope-tagged FHK near the kinetoplast.** A, B) BF expressing FHK-V5 and co-stained with DAPI, anti-V5 antibodies, and either anti-PIFTC3 (A) or anti-BiP (B). Enlargement shows region around kDNA. The cell in (A) has duplicated both its nucleus (N) and kDNA (k) and is undergoing cytokinesis as evidenced by the presence of two flagella (F). The cell in panel B has an elongated, slightly V-shaped kinetoplast, indicating it is undergoing kinetoplast division and is in nuclear S phase. Note that a point of FHK-staining occurs at either pole of the elongated kDNA. C) PF expressing FHK-HA were incubated with biotinylated-tomato lectin (TL) at 4° and fixed prior to staining with DAPI, avidin and anti-HA. Enlargement shows region around kDNA. D) Whole cell and enlarged serial planes (numbered) of a DAPI-stained PF cell expressing FHK-HA. All images (except enlargements) are at the same magnification. Bar = 5 μm.

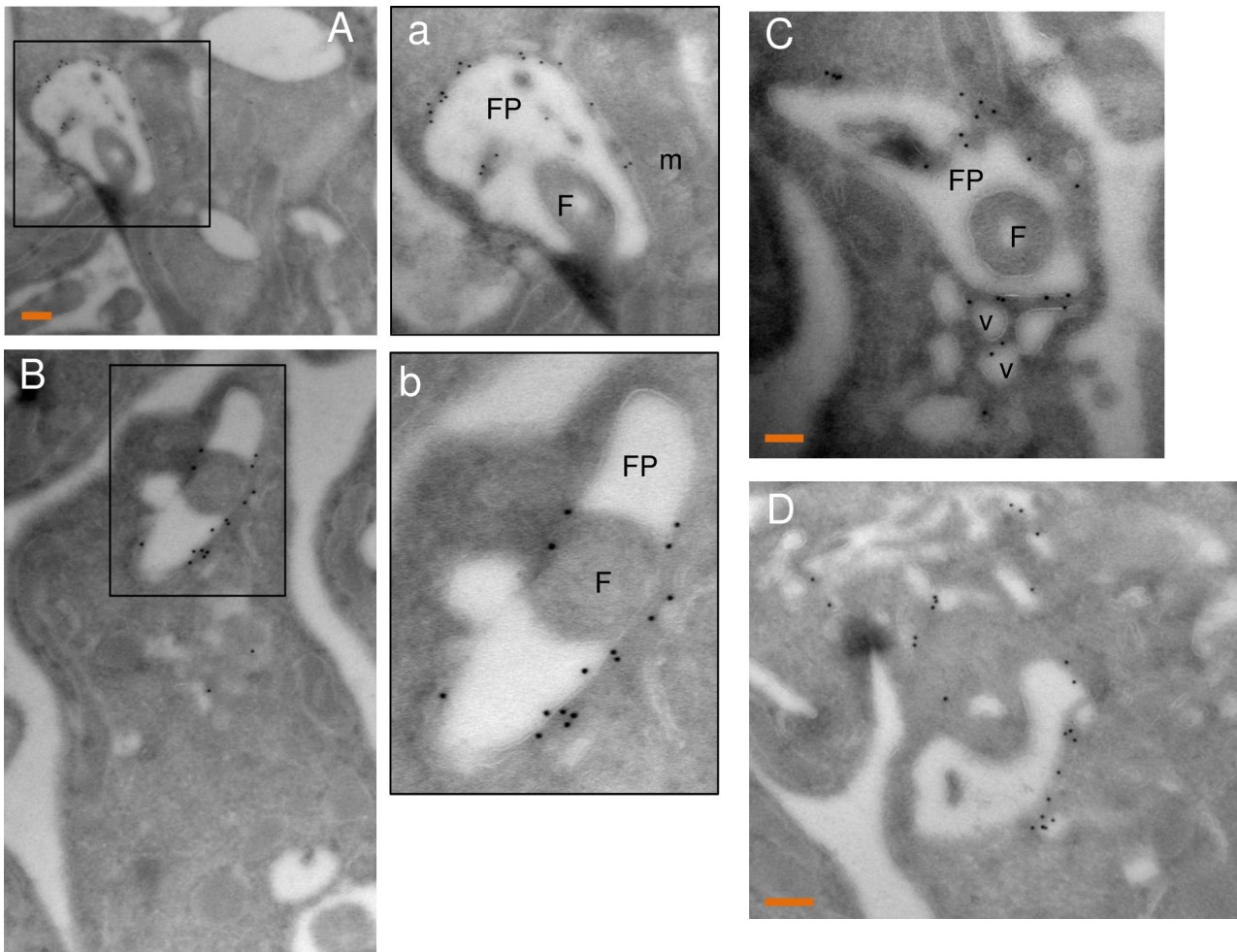

**Fig 9. Localization of FHK-HA to the flagellar pocket of procyclic forms by immunoelectron microscopy.** Procyclic form parasites expressing FHK C-terminally tagged with HA epitopes were stained with anti-HA revealed by protein-A gold 10 nm particles. A) The gold staining is on the flagellar pocket (FP) membrane but absent from the flagellum (F) and mitochondrion (m). Enlargement of the marked region is shown in panel a. B) Staining is predominantly at the flagellar pocket. Enlargement of the marked region is shown in panel b. C) The staining is sometimes seen on vesicles close to the FP. D) Occasional parasites showed staining outside the flagellar pocket. Bars, 100 nm.

activity, immunoprecipitates from PF expressing FHK-HA, as well as the untransfected, parental single marker line, were assayed as described for MEKK1. As shown by phosphorimaging (Fig 10B, top) and western blot (Fig 10B, bottom), MBP was phosphorylated in the sample containing FHK-HA, but not the untransfected control, providing evidence of FHK catalytic activity. As with MEKK1, it is formally possible, albeit unlikely, that tagged FHK is inactive but interacts with an active kinase responsible for the detected activity.

## Discussion

### A plethora of transmembrane domains

In contrast to metazoan kinomes, which have large numbers of TMD PKs, protozoan kinomes vary widely in their representation of TMD PKs. At one extreme, the *Giardia lamblia* genome encodes only two predicted transmembrane PKs (two other PKs have predicted TMDs unlikely to be authentic since they punctuate the catalytic domain). At the other extreme is

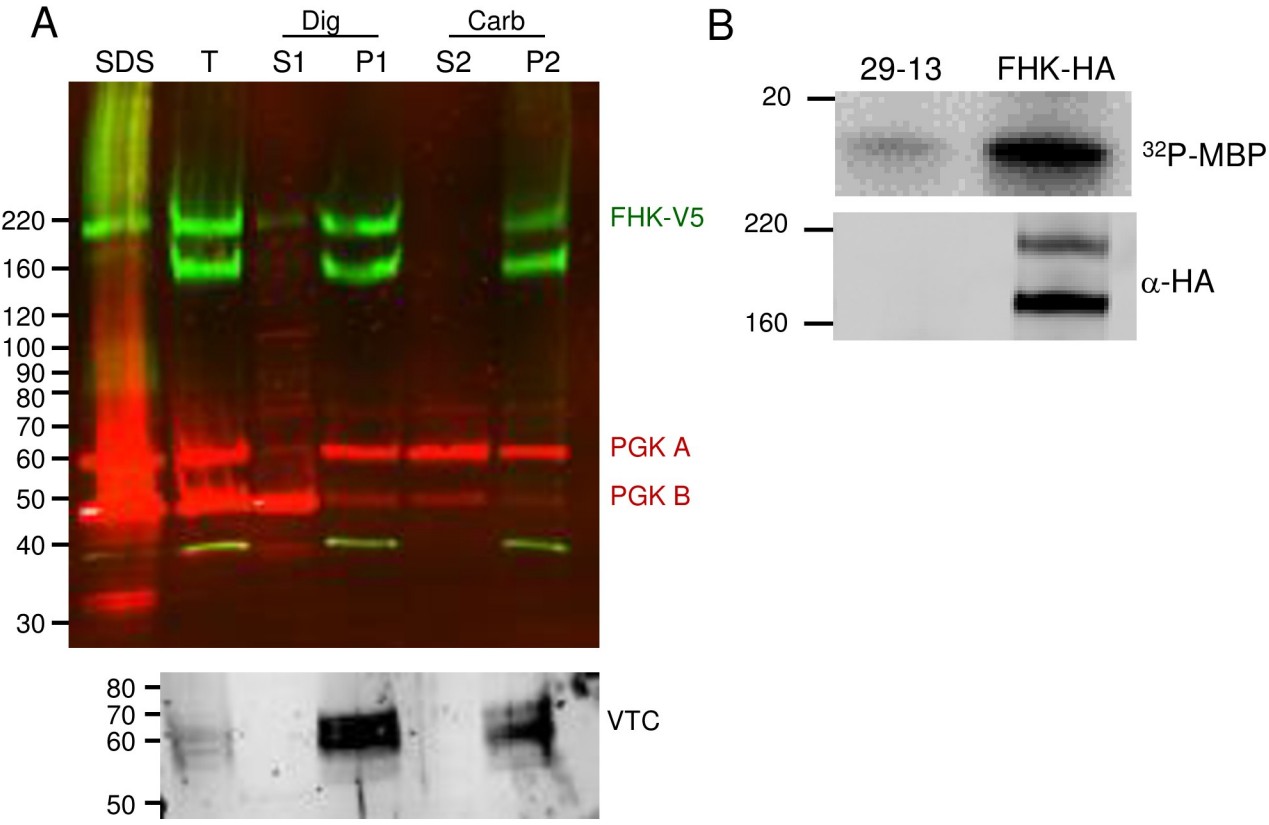

**Fig 10. FHK is an integral membrane protein with protein kinase activity.** A) Cell fractionation and immunoblotting were as described in Fig 7. PF cells expressing FFK-V5 were permeabilized and fractionated. The pellet representing organelles was treated with carbonate and further fractionated After separation by SDS-PAGE and blotting, the blots were probed with mouse anti-HA (green) and rabbit anti-phosphoglycerate kinase (red) which detects the 56 kDa glycosomal form (PGKA) and the 47 kDa cytosolic isoform (PGKB). A parallel blot was also incubated with antibody to the acidocalcisome membrane protein vacuolar phosphatase (VTC). SDS, SDS lysate; T, total after incubation with digitonin; S1, digitonin supernatant; P1, digitonin pellet; S2, carbonate supernatant; P2, carbonate pellet. Lanes contain fractions from $3 \times 10^6$ cells. B) Kinase activity as described in Fig 9. FHK-HA was immunoprecipitated from PF and incubated with $\gamma$ $^{32}$P-ATP and MBP. A negative control immunoprecipitation was performed using the untransfected parental line 29–13. Top: phosphorimaging demonstrates phosphorylation of MBP. Bottom: the blot was probed with anti-HA mAb demonstrating the pulldown of the tagged protein.

*Entamoeba histolytica* with more than 80 predicted TMD PKs, most of which are related at the sequence level and some of which are likely non-catalytic [50]. Like *E. histolytica*, *Plasmodium falciparum* has evolved a unique set of PKs, the FIKK kinases, that comprise the majority of the parasite's 17 PKs with predicted TMDs. The FIKK kinases bear the signature sequence for export out of the parasite [51], and at least some of them are important in remodeling the host erythrocyte [52, 53]. Like *P. falciparum*, trypanosomatids fall between extremes of *G. lamblia* and *E. histolytica*, with a paucity of PKs that bear TMDs compared to metazoans. These trypanosomatid PKs are classified firmly with serine/threonine kinases as opposed to tyrosine kinases characteristic of transmembrane PKs of animals [7]. Just as interesting, most of these trypanosomatid PKs are predicted to have multiple membrane spanning domains, a feature that is extremely rare in eukaryotes. To our knowledge, a single ePK domain protein in eukaryotes has been biochemically demonstrated to have more than one TMD, lemur tail kinase 2 [54], but other members of that family of PKs are also predicted to have two TMDs [55]. Interestingly, in perusing the *E. histolytica* genome database, we noted a few PKs with two likely TMDs. Another interesting observation is that many bacterial histidine kinases with several TMDs have been identified in genome-wide bioinformatic analyses [56]. These include the

5-TMD bearing *Staphylococcus aureus* LytS histidine kinase which acts as a sensor of membrane potential [57]. Another example is *Bacillus subtilis* DesK, in which TMDs are involved in signaling temperature-induced changes in membrane fluidity [58]. Thus, TMDs can modulate kinase activity in response to environmental changes even when no extracellular ligand binding domains are present. It will be interesting to determine if any of the TMDS in the *T. brucei* PKs examined here have such effects.

## Endomembrane kinases

We observed that five of the tagged PKs localized to the endomembrane system, most likely the ER, but cannot exclude the possibility that some of these proteins reach the plasma membrane. Typically, this can be explored by antibody staining of live cells rather than permeabilized cells. However, antibodies cannot penetrate the VSG surface coat of live bloodstream forms, so proteins that lie under the coat cannot be visualized prior to fixation and permeabilization. The mechanisms by which these proteins remain in the ER is unclear. Sequences specifying ER retention/retrieval of membrane proteins are often cytosolic, such as KKXX motifs near the C-terminus (KKXX) or RXR motifs. Additionally, ER retrieval can be achieved via specific residues within TMDs which interact with a retrieval protein RER1 [59] (a homologue is present in *T. brucei*). In many eukaryotes, two protein kinases localized to the endoplasmic reticulum, IRE1 and the eIF2 alpha kinase PERK play prominent roles in stress responses [60]. While these PKs colocalized with BiP, indicating that the PKs are entering the endomembrane system, it does not indicate that their destination is not further in the endomembrane system either in the Golgi, lysosome or endosomes. Notably, Tb927.3.3150 had additional punctate localization proximal to the kinetoplast in the region of the cell that contains the endosomes, while Tb927.9.12400 had posterior staining proximal to the nucleus.

## Flagellar and flagellar pocket localization and function

The localization of MEKK1 and FHK to the flagellum and flagellar pocket respectively raises the question of what biochemical processes they may regulate. Initiation of stumpy development appears to be triggered by the uptake of oligopeptides by the transporter GPR89 [61]. MEKK1 acts early in the developmental pathway, which involves multiple additional kinases [8]. How the oligopeptide signal intersects with MEKK1 is not clear given that GPR89 is distributed across the plasma membrane and is not specifically localized to, or perhaps not present at all, at the flagellar membrane. However, other molecules that may be involved in development are also localized to the flagella, such as certain adenylate cyclases and phosphodiesterases as well as a calcium-activated channel [62].

In trypanosomatids, the flagellar pocket is the site of the essential processes of exocytosis and endocytosis. The localization of FHK to the flagellar pocket could suggest a function in these pathways. However, RNAi data suggests that FHK is dispensable in BF [9, 63] (also corroborated by our preliminary data), so it is unlikely that it functions as a global regulator of these essential processes. Instead, its function may be focused on specific conditions (e.g., stress response) or certain pathways within the broader categories of exocytosis, endocytosis and recycling of membrane molecules. The flagellar pocket bears many distinct proteins, so another alternative is that FHK is positioned to sense interaction with transmembrane proteins or even biophysical changes in this distinct membrane domain, such as those sensed by LytS and DesK.

## Signaling domains of likely bacterial origin

There are numerous genes in trypanosomatids, especially those encoding metabolic enzymes, which encode proteins more closely related to bacterial than to eukaryotic homologues,

providing evidence of lateral gene transfer. Our HHpred analysis showed that two *T. brucei* PKs have regions predicted at >95% probability to fold similarly to bacterial signaling modules. FHK has a region resembling ligand-sensing domains found on chemotaxis proteins and certain histidine kinases, but not on eukaryotic proteins. DAK has a cassette bearing four regions common to hybrid histidine kinases: a sensor, a dimerization/phosphoacceptor, a catalytic domain and a receiver domain. These modular structures are conserved in genomes across trypanosomatids and in the bodonid *Bodo saltans*, ruling out an artifactual explanation and arguing for a functional and possibly regulatory role. While the combination of domains seen in DAK is unusual, Uniprot searches do identify predicted proteins that possess both an ePK domain and a histidine kinase cassette. Most of these predicted proteins are bacterial, but some are found in eukaryotes (predominantly fungi). Those in eukaryotes almost exclusively have hybrid type histidine kinase domains (i.e, with a receiver domain on the same molecule) as is seen on DAK. Few (if any) of these histidine-ePK kinases have been explored functionally to understand how the domains interact and influence the signaling properties of the protein. DAK's potential for histidine kinase activity that phosphorylates the DHp region appears low since the conserved histidine is mutated. However, the aspartic acid in the receiver domain that accepts the phosphate is still present. Clearly additional work will need to be done to understand the roles of the two ancestral kinase regions on the kinase activity and ultimately function of DAK [44].

## Supporting information

**S1 Fig. RDK1 structural similarities identified by HHpred.** The C-terminal PK domain is not shown. Red ovals–transmembrane domains predicted by TMHMM and CCTOP. Each bar below the gene model represents the top hits by HHpred. The color of the arrows mark descriptions matching the hits. The description includes the PDB number and chain designation (Hit), a description of the PDB entry including related PDB entries (Name), the probability of the hit based on the Hidden Markov Model (Probability), the probability of the match in an unrelated database (E-value), score for the secondary structure prediction (SS), number of amino acids aligned (Col), and the total length of the target in PDB (Target Length). The documentation for HHpred considers Probability the most important criterion with positive hits meeting at least one of these criteria: having a score >95% or having a score >50% and making reasonable biological sense [33]. *T. cruzi* and *L. major* orthologues have the same predicted folds.
(PDF)

**S2 Fig. Western blot showing expression of epitope-tagged PKs in *T. brucei* bloodstream forms.** Parasites were grown in the absence of Tet or in the presence of Tet to induce the V-5 tagged PK. Two independent clonal isolates are shown for FHK. Clone 2 was used for the localization data shown in Fig 8. The migration of each tagged protein is marked. The predicted molecular weights for each protein are provided in S1 Table.
(PDF)

**S1 Table. Primers used.**
(PDF)

**S2 Table. TriTryp orthologues—transmembrane domains, signal sequences, and additional domains.**
(PDF)

**S1 Raw images.**
(PDF)

## Acknowledgments

We wish to thank David Horn for plasmid pT7-GFP and George Cross for plasmid pLEW100v5-BSD. We additionally thank Jay Bangs (BiP), Elisabetta Ullu (PIFCT3) and Roberto Docampo (TcCat and TbVTC) for the gifts of the indicated antibodies. We thank the Yale Microscopy Facility (K. Zichichi) for performing immunoEM.

## Author Contributions

**Conceptualization:** Bryan C. Jensen, Marilyn Parsons.

**Formal analysis:** Bryan C. Jensen, Marilyn Parsons.

**Funding acquisition:** Marilyn Parsons.

**Investigation:** Bryan C. Jensen, Pashmi Vaney, John Flaspohler, Isabelle Coppens.

**Methodology:** Bryan C. Jensen, Marilyn Parsons.

**Project administration:** Marilyn Parsons.

**Supervision:** Bryan C. Jensen, Marilyn Parsons.

**Writing – original draft:** Bryan C. Jensen, Marilyn Parsons.

**Writing – review & editing:** Bryan C. Jensen, Pashmi Vaney, John Flaspohler, Isabelle Coppens, Marilyn Parsons.

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
