## [Decision Letter · Decision Letter 0]

4 Sep 2021

PONE-D-21-23128

Unusual Features of the Membrane Kinome of *Trypanosoma brucei*

PLOS ONE

Dear Dr. Jensen,

Thank you for submitting your manuscript to PLOS ONE. After careful consideration, we feel that it has merit but does not fully meet PLOS ONE’s publication criteria as it currently stands. Therefore, we invite you to submit a revised version of the manuscript that addresses the points raised during the review process.

Please be sure to address all reviewer comments.

We look forward to receiving your revised manuscript.

Kind regards,

Ben L. Kelly, Ph.D.

Academic Editor

PLOS ONE

“This work was supported in part by a grant 5R21AI101424 from the National Institutes

of Health and by Seattle Children’s Research Institute to MP.”

Reviewers' comments:

Reviewer's Responses to Questions

**Comments to the Author**

1. Is the manuscript technically sound, and do the data support the conclusions?

Reviewer #1: Yes

Reviewer #2: Yes

2. Has the statistical analysis been performed appropriately and rigorously? 

Reviewer #1: N/A

Reviewer #2: Yes

3. Have the authors made all data underlying the findings in their manuscript fully available?

Reviewer #1: Yes

Reviewer #2: Yes

4. Is the manuscript presented in an intelligible fashion and written in standard English?

Reviewer #1: Yes

Reviewer #2: Yes

5. Review Comments to the Author

Reviewer #1: This paper identifies a number of genes in T. brucei that may operate as membrane bound protein kinases using in silico protein prediction methods. The authors go on to localize a subset of them using immunofluorescence and EM. Of the membrane bound protein kinases examined, 5 localized to the ER and 2 localized to the flagellum or flagellar pocket. The data on localization is convincing. Other reviewers have pointed out that while the study makes a good beginning at the study of these proteins, it lacks functional data that can point to any biological role for the proteins (leaving aside MEKK1, which has already been studied). I agree that the study has not progressed very far with respect to biological function of the kinases under study. However, I imagine that the localization of these proteins might be helpful for other researchers interested in the study of kinases in T. brucei. Given the limited amount of functional information in the paper, one might imagine that this group may be ready to focus their effort somewhere else. In my view, it would be a shame to leave their localization efforts unpublished, as the localization serves as a nice jumping off point for further study that might be helpful for other groups. Thus, PLoS One, with its emphasis on quality of data rather than impact, is a good home for this slightly limited but nonetheless valuable body of work. The authors have nicely addressed concerns brought up by other reviewers, and I have only a few minor comments to add.

1. The title is a little vague, and doesn’t really convey the main result of the paper, which is the localization of a subset of the membrane kinases. Perhaps something along the lines of

Identification and Localization of 7 members of the Membrane kinome in Trypanosoma brucei

might more accurately convey the main findings of the paper. The authors will likely be better at wordsmithing it than me.

2. The second to last paragraph of the Intro section introduces our main membrane kinome players and what’s known about them in the literature. It might be helpful to have that information in a small table that can easily be referred to as the reader continues. Alternatively the information could be added easily to Table 1 with a one-liner comment column.

3. In line 93 the authors mention that the monomorphic BF cannot differentiate to PFs. It might be more accurate to state that monomorphs can’t be differentiated to stumpy forms. Procyclics can be produced from monomorphs using temperature and acid signals in the form of cis-aconitate, but true stumpy populations can’t be generated.

4. In lines 312-316 the authors mention that their Dox induced tagged lines might produce overexpressed proteins. Although the authors have nicely demonstrated that transcript levels don’t necessarily correlate perfectly with protein levels, a q-PCR experiment using primers that don’t distinguish between the introduced tagged genes and the endogenous genes could give some indication as to the degree of overexpression that might be present here.

5. Thinking along the lines of point 4 above, the authors could endogenously tag just one of their ER kinases to show that overexpressing these proteins does not lead to changes in their localization. Since MEKK1 has already been localized to the flagellar pocket by the tryptag team this endogenous tagging is not so important.

6. In Figure 1, it would be helpful to put a small legend within the figure panel that describes the red stripes, red arrowheads, asterisks, and the red X in addition to having that information in the legend.

7. Figure 2. Do you need the bottom two rows of red bars? Do they correspond to entries 4 and 5 in the bottom table? I wasn’t clear what these rows were adding or why they were there.

8. Line 317 the period in the subject heading is underlined.

9. Line 320 the comma after the word BiP should be a period.

10. Line 327 Pearson;s correlation should read Pearson’s correlation.

11. For Figure 4 please make the green arrows white. The green arrows are really hard to see on these images.

12. In Figures 5, 6, and 8 the authors are switching back and forth between using bloodstream and procyclic cells. I don’t think there is a problem with this, but would it be possible to put the reasoning behind it into the text? Does it have to do with expression levels? Or something else?

13. For Figure 5, it would be more clear if the authors annotated the image labels to help out the people a little outside this field. V5 could be labeled as BF MEKK1-V5. TL could be labeled as TL(flagellar pocket). PIFTC3 could be labeled as PIFTC3 (flagellar transport) and YL1/2 could be labeled as YL1/2 (basal bodies).

14. Figure 5. I may have missed it, but I didn’t see a mention of the result using the basal body YL1/2 markers in the text, although it’s present in the figure legend. Could a more explicit statement of this result be included in the main text?

15. In Figure 4, the authors performed a correlation analysis for the overlap between their ER localized kinases and the ER marker BiP. Would it be possible to perform this analysis for the overlap of MEKK1 and the flagellar pocket shown in the top panels of Figure 5? Same goes for Fig 8C, would it be possible to perform the correlation analysis for the overlap of FHK with the flagellar pocket?

16. In Figure 6 could the legend title be changed to Localization of PF MEKK1-HA to the flagellum and flagellar pocket….This would make it more consistent with the label of Figure 5.

17. This isn’t a major issue, but the blots shown in Figures 7 and 10 in the bottom panels and in panel B are missing size markers.

18. Figure 8 would be more clear if the image was labeled as BF FHK-V5 and PF FHK-HA. Same comment as for TL in item 13 above.

19. In the legend for Fig 10 please rewrite the info in Fig 7 instead of referencing Fig 7 so that the Figure legend and can stand on its own.

20. Line 518 contains an extra word ‘be’

21. Line 527 there is a missing ‘a’ between DAK has and cassette

22. The blot in S2 Figure for FHK looks a bit wonky. Was there a bubble in the vicinity of the -Tet sample? Would it be too much trouble to do a quick redo? This is not mission critical, but might make the inducible expression more convincing for FHK since it’s hard to see what’s going on in the -Dox lane.

23. The authors mention in response to a previous reviewer that ‘It is formally possible, albeit unlikely, that the tagged kinase is inactive but interacts with an active kinase responsible for the detected activity.’ A one liner to that effect in the Discussion section would not be amiss.

24. Please define TMHMM the first time it’s used.

Reviewer #2: In the manuscript, “Unusual Features of the Membrane Kinome of Trypanosoma brucei”, Jensen and colleagues performed various in silico analyses to identify ten predicted protein kinases in T. brucei with one or more putative membrane domains. Additional in silico analyses identified domains with primary amino acid sequence homology, or in some cases predicted 3D structural similarity, to motifs in other proteins that could help provide insights into the functions of these (mostly) uncharacterized kinases. Of particular note, RDK1 and DAK (the name proposed by the authors) contain both eukaryotic protein kinase domains and several motifs resembling bacterial two component regulators, including sensor, receiver and histidine kinase domains (though histidine kinase activity is unlikely to be conserved). As sensing and signaling in these parasites is poorly understood, elucidating the biological roles of these “hybrid” kinases could prove quite valuable. The authors used their previously published ribosome profiling data to provide insights into the relative expression of the putative kinases in slender BSF and PF parasites. The subcellular localization of ectopically expressed epitope-tagged versions of seven putative kinases was determined via immunogold electron microscopy and/or immunofluorescence. The majority localized to endomembranes, most likely the ER. The remaining two, MEKK1 and FHK, localized to the flagellum and/or flagellar pocket and the authors subsequently demonstrated that both proteins have kinase activity in vitro.

This manuscript has been previously reviewed following submission to PLoS Neglected Tropical Diseases. It would appear that the main issue precluding it from publication in PLoS NTDs was the perception that the work lacked sufficient impact/novelty, which isn’t an issue with PLoS ONE. The authors have adequately addressed the issues from the previous reviewers’ comments, and in general the work is carefully done and well presented. In particular, the inclusion of the Pearson correlation coefficients strengthens the colocalization data. While not groundbreaking, overall the manuscript provides useful information about an interesting class of protein kinases in kinetoplastids. Other than a few formatting errors that can be corrected in the editing process (see below), I have no substantive criticisms of the work. I recommend that the manuscript be accepted for publication.

Formatting issues:

Line 316: should read, “PKs are most likely to be over-expressed.”

Line 320: there should be a period between “BiP” and “Staining”, not a comma

6. PLOS authors have the option to publish the peer review history of their article (what does this mean?). If published, this will include your full peer review and any attached files.

Reviewer #1: No

Reviewer #2: No

---

## [Author Response · Author response to Decision Letter 0]

4 Oct 2021

Reviewer #1

This paper identifies a number of genes in T. brucei that may operate as membrane bound protein kinases using in silico protein prediction methods. The authors go on to localize a subset of them using immunofluorescence and EM. Of the membrane bound protein kinases examined, 5 localized to the ER and 2 localized to the flagellum or flagellar pocket. The data on localization is convincing. Other reviewers have pointed out that while the study makes a good beginning at the study of these proteins, it lacks functional data that can point to any biological role for the proteins (leaving aside MEKK1, which has already been studied). I agree that the study has not progressed very far with respect to biological function of the kinases under study. However, I imagine that the localization of these proteins might be helpful for other researchers interested in the study of kinases in T. brucei. Given the limited amount of functional information in the paper, one might imagine that this group may be ready to focus their effort somewhere else. In my view, it would be a shame to leave their localization efforts unpublished, as the localization serves as a nice jumping off point for further study that might be helpful for other groups. Thus, PLoS One, with its emphasis on quality of data rather than impact, is a good home for this slightly limited but nonetheless valuable body of work. The authors have nicely addressed concerns brought up by other reviewers, and I have only a few minor comments to add. 

1. The title is a little vague, and doesn’t really convey the main result of the paper, which is the localization of a subset of the membrane kinases. Perhaps something along the lines of 

Identification and Localization of 7 members of the Membrane kinome in Trypanosoma brucei 

might more accurately convey the main findings of the paper. The authors will likely be better at wordsmithing it than me. 

We have modified the title to include the word localization.

2. The second to last paragraph of the Intro section introduces our main membrane kinome players and what’s known about them in the literature. It might be helpful to have that information in a small table that can easily be referred to as the reader continues. Alternatively the information could be added easily to Table 1 with a one-liner comment column.

This information has been added to Table 1

3. In line 93 the authors mention that the monomorphic BF cannot differentiate to PFs. It might be more accurate to state that monomorphs can’t be differentiated to stumpy forms. Procyclics can be produced from monomorphs using temperature and acid signals in the form of cis-aconitate, but true stumpy populations can’t be generated.

We clarify that the clone that we use cannot differentiate to PF, and we have removed the generalized statement. 

4. In lines 312-316 the authors mention that their Dox induced tagged lines might produce overexpressed proteins. Although the authors have nicely demonstrated that transcript levels don’t necessarily correlate perfectly with protein levels, a q-PCR experiment using primers that don’t distinguish between the introduced tagged genes and the endogenous genes could give some indication as to the degree of overexpression that might be present here. 

Since it is likely that the tagged proteins are overexpressed, simply showing that they are does not add much to the interpretation. 

5. Thinking along the lines of point 4 above, the authors could endogenously tag just one of their ER kinases to show that overexpressing these proteins does not lead to changes in their localization. Since MEKK1 has already been localized to the flagellar pocket by the tryptag team this endogenous tagging is not so important.

We agree that this could be useful, although it is a low priority given the TrypTag results. 

6. In Figure 1, it would be helpful to put a small legend within the figure panel that describes the red stripes, red arrowheads, asterisks, and the red X in addition to having that information in the legend. 

Done

7. Figure 2. Do you need the bottom two rows of red bars? Do they correspond to entries 4 and 5 in the bottom table? I wasn’t clear what these rows were adding or why they were there. 

The bottom two rows were included since the examples further validate the high probability hits seen in the rows above. 

8. Line 317 the period in the subject heading is underlined.

Corrected. 

9. Line 320 the comma after the word BiP should be a period.

Corrected.

10. Line 327 Pearson;s correlation should read Pearson’s correlation. 

Corrected.

11. For Figure 4 please make the green arrows white. The green arrows are really hard to see on these images. 

We like having the arrows green because corresponds to the color of the signal, However, we have significantly lightened the shade of green to increase contrast.

12. In Figures 5, 6, and 8 the authors are switching back and forth between using bloodstream and procyclic cells. I don’t think there is a problem with this, but would it be possible to put the reasoning behind it into the text? Does it have to do with expression levels? Or something else? 

We have added language to further clarify when PF are used and explain the rationale for using PF for specific experiments. We have also expanded Fig 5 to show the same localization of the MEKK1 in PF, providing further justification that the use of PF is appropriate. 

13. For Figure 5, it would be more clear if the authors annotated the image labels to help out the people a little outside this field. V5 could be labeled as BF MEKK1-V5. TL could be labeled as TL(flagellar pocket). PIFTC3 could be labeled as PIFTC3 (flagellar transport) and YL1/2 could be labeled as YL1/2 (basal bodies). 

Labels have been added to the figures. However, the potassium channel was not labeled as flagellar attachment zone, since the localization of this protein has only been shown in T. cruzi and not in T. brucei. However, we note the T. cruzi localization in the legend.

14. Figure 5. I may have missed it, but I didn’t see a mention of the result using the basal body YL1/2 markers in the text, although it’s present in the figure legend. Could a more explicit statement of this result be included in the main text? 

Done

15. In Figure 4, the authors performed a correlation analysis for the overlap between their ER localized kinases and the ER marker BiP. Would it be possible to perform this analysis for the overlap of MEKK1 and the flagellar pocket shown in the top panels of Figure 5? Same goes for Fig 8C, would it be possible to perform the correlation analysis for the overlap of FHK with the flagellar pocket?

The immumo EM shows definitive localization of these two proteins to the flagellar pocket and adjacent structures. This is a more direct assement of localization than using correlation analysis of IFA images.

16. In Figure 6 could the legend title be changed to Localization of PF MEKK1-HA to the flagellum and flagellar pocket….This would make it more consistent with the label of Figure 5. 

Done

17. This isn’t a major issue, but the blots shown in Figures 7 and 10 in the bottom panels and in panel B are missing size markers. 

Size markers have been added to the panels.

18. Figure 8 would be more clear if the image was labeled as BF FHK-V5 and PF FHK-HA. Same comment as for TL in item 13 above. 

Done

19. In the legend for Fig 10 please rewrite the info in Fig 7 instead of referencing Fig 7 so that the Figure legend and can stand on its own. 

We have updated the legend for Fig 10 so that it stands on its own.

20. Line 518 contains an extra word ‘be’

Corrected.

21. Line 527 there is a missing ‘a’ between DAK has and cassette

Corrected

22. The blot in S2 Figure for FHK looks a bit wonky. Was there a bubble in the vicinity of the -Tet sample? Would it be too much trouble to do a quick redo? This is not mission critical, but might make the inducible expression more convincing for FHK since it’s hard to see what’s going on in the -Dox lane.

We have replaced this blot with a different blot that is not wonky.

23. The authors mention in response to a previous reviewer that ‘It is formally possible, albeit unlikely, that the tagged kinase is inactive but interacts with an active kinase responsible for the detected activity.’ A one liner to that effect in the Discussion section would not be amiss.

This comment was added to the Results section for the specific experiments as there was no appropriate place in the Discussion. 

24. Please define TMHMM the first time it’s used. 

Done: Transmembrane Hidden Markov Model.

Reviewer #2: 

In the manuscript, “Unusual Features of the Membrane Kinome of Trypanosoma brucei”, Jensen and colleagues performed various in silico analyses to identify ten predicted protein kinases in T. brucei with one or more putative membrane domains. Additional in silico analyses identified domains with primary amino acid sequence homology, or in some cases predicted 3D structural similarity, to motifs in other proteins that could help provide insights into the functions of these (mostly) uncharacterized kinases. Of particular note, RDK1 and DAK (the name proposed by the authors) contain both eukaryotic protein kinase domains and several motifs resembling bacterial two component regulators, including sensor, receiver and histidine kinase domains (though histidine kinase activity is unlikely to be conserved). As sensing and signaling in these parasites is poorly understood, elucidating the biological roles of these “hybrid” kinases could prove quite valuable. The authors used their previously published ribosome profiling data to provide insights into the relative expression of the putative kinases in slender BSF and PF parasites. The subcellular localization of ectopically expressed epitope-tagged versions of seven putative kinases was determined via immunogold electron microscopy and/or immunofluorescence. The majority localized to endomembranes, most likely the ER. The remaining two, MEKK1 and FHK, localized to the flagellum and/or flagellar pocket and the authors subsequently demonstrated that both proteins have kinase activity in vitro.

This manuscript has been previously reviewed following submission to PLoS Neglected Tropical Diseases. It would appear that the main issue precluding it from publication in PLoS NTDs was the perception that the work lacked sufficient impact/novelty, which isn’t an issue with PLoS ONE. The authors have adequately addressed the issues from the previous reviewers’ comments, and in general the work is carefully done and well presented. In particular, the inclusion of the Pearson correlation coefficients strengthens the colocalization data. While not groundbreaking, overall the manuscript provides useful information about an interesting class of protein kinases in kinetoplastids. Other than a few formatting errors that can be corrected in the editing process (see below), I have no substantive criticisms of the work. I recommend that the manuscript be accepted for publication.

Formatting issues:

Line 316: should read, “PKs are most likely to be over-expressed.”

Corrected. 

Line 320: there should be a period between “BiP” and “Staining”, not a comma

 Corrected.

---

## [Decision Letter · Decision Letter 1]

6 Oct 2021

Unusual Features and localization of the Membrane Kinome of *Trypanosoma brucei*

PONE-D-21-23128R1

Dear Dr. Jensen,

We’re pleased to inform you that your manuscript has been judged scientifically suitable for publication and will be formally accepted for publication once it meets all outstanding technical requirements.

Kind regards,

Ben L. Kelly, Ph.D.

Academic Editor

PLOS ONE

Additional Editor Comments (optional):

Reviewers' comments:

Reviewer's Responses to Questions

**Comments to the Author**

1. If the authors have adequately addressed your comments raised in a previous round of review and you feel that this manuscript is now acceptable for publication, you may indicate that here to bypass the “Comments to the Author” section, enter your conflict of interest statement in the “Confidential to Editor” section, and submit your "Accept" recommendation.

Reviewer #1: All comments have been addressed

2. Is the manuscript technically sound, and do the data support the conclusions?

Reviewer #1: Yes

3. Has the statistical analysis been performed appropriately and rigorously? 

Reviewer #1: Yes

4. Have the authors made all data underlying the findings in their manuscript fully available?

Reviewer #1: Yes

5. Is the manuscript presented in an intelligible fashion and written in standard English?

Reviewer #1: Yes

6. Review Comments to the Author

Reviewer #1: All my comments have been addressed, thank you! I look forward to seeing this paper published.

7. PLOS authors have the option to publish the peer review history of their article (what does this mean?). If published, this will include your full peer review and any attached files.

Reviewer #1: No

---

## [Editor Report · Acceptance letter]

8 Oct 2021

PONE-D-21-23128R1 

Unusual Features and localization of the Membrane Kinome of *Trypanosoma brucei*

Dear Dr. Jensen:

I'm pleased to inform you that your manuscript has been deemed suitable for publication in PLOS ONE. Congratulations! Your manuscript is now with our production department. 

Kind regards, 

on behalf of

Dr. Ben L. Kelly 

Academic Editor

PLOS ONE